# Understanding the Anticancer Properties of Honey

**DOI:** 10.3390/ijms252111724

**Published:** 2024-10-31

**Authors:** Simona Martinotti, Gregorio Bonsignore, Elia Ranzato

**Affiliations:** Dipartimento di Scienze e Innovazione Tecnologica (DiSIT), University of Piemonte Orientale, 15121 Alessandria, Italy; simona.martinotti@uniupo.it (S.M.); gregorio.bonsignore@uniupo.it (G.B.)

**Keywords:** anti-proliferative effects, cancer cells, honey, oxidative stress

## Abstract

Uncontrolled cell growth that possesses the capacity to exhibit malignant behavior is referred to as cancer. The cytotoxic drugs used to fight cancer are associated with several adverse effects and are not always readily available or affordable, especially in developing countries. These issues are in addition to the shortcomings of the current cancer treatment regimen. According to growing research, honey is not cytotoxic to normal cells but is highly and particularly cytotoxic to tumor cells, suggesting that honey may display anticancer effects. Research has shown that honey affects a number of cell signaling pathways; however, at the moment, the precise method is not completely known.

## 1. Introduction

Cancer is the term for unchecked cell development that has the potential to become malignant. It ranks among the world’s main causes of death [1]. Over 35 million new cancer cases are predicted in 2050, a 77% increase from the estimated 20 million cases in 2022 [2].

This rising prevalence can be attributed to several factors, such as an aging population, the adoption of a Western lifestyle in many developing countries, increased awareness, and improved cancer screening and diagnosis.

In addition to the shortcomings of the current cancer treatment regimen, which consists of surgery, chemotherapy, and radiation therapy, the use of cytotoxic drugs is linked to a number of unfavorable side effects, and these drugs are not always readily affordable or accessible, particularly in developing nations [3]. Because of this, a sizable segment of the populace favors using complementary and alternative medicine (CAM). Compared to synthetic or conventional medications, CAM offers several benefits, including less side effects, availability, and price [4,5]. Growing public interest in the use of CAM can be linked to the corpus of research showing some benefits of CAM. Conventional medicine is based on techniques that have been shown via well-planned trials and research to be secure and efficient. However, a lot of CAM therapies still lack reliable research upon which to make well-informed choices. Many CAM treatments have unproven risks and potential benefits.

In general, CAM refers to a variety of therapeutic modalities (medical goods and procedures) that do not form an essential part of mainstream medicine. These modalities may include nutritional supplements, herbal treatments, naturally derived goods, and even lifestyle modifications [4].

Cancer patients now use and are more aware of complementary and alternative medicine [6]. One such product that has grown to be a crucial part of complementary and alternative medicine is honey, a natural material made by honeybees from nectar [7]. However, numerous questions remain unresolved. To better understand honey’s beneficial effects on cancer, more research is required. Humans may not experience what is observed in animal experiments or cell cultures. To confirm the legitimacy of honey, either by itself or in conjunction with adjuvant therapy, prospective randomized controlled clinical trials are required.

## 2. Honey Composition

Due to the various components’ actions and the notable variations in these chemicals’ concentrations between honeys, the therapeutic potential of honey is highly complex.

Moreover, a number of factors, including exposure to heat and light, may affect honey’s therapeutic properties.

Honey is a naturally occurring product that is either semi-solid or heavily soaked. The bees use nectar, which is mostly made up of sugars with different levels of moisture, mineral contents, enzymes, and phytochemicals depending on the floral source, as its primary source of carbohydrates. After being carried back to the hive by foragers, nectar is placed into the honeycomb’s wax cells, where it undergoes ripening, the physicochemical conversion to honey. By releasing different enzymes into the nectar from their hypopharyngeal glands, worker bees continue to control the forming honey during this phase. These enzymes have the ability to break down the components of the nectar. The most significant is invertase, which breaks down sucrose into fructose and glucose to produce a stable, dense, and extremely active food. Additional enzymes include diastase, which breaks down starch and dextrins into smaller carbohydrates and is believed to be involved in the digestion of pollen, the primary source of protein for honey bees, and protease, which breaks down proteins and polypeptides to produce smaller peptides that may affect the quality and nutritional value of honey [8]. Active and passive evaporation reduces the nectar’s moisture content while this biochemical reaction is occurring, increasing its resistance to microbial deterioration and extending its shelf life. Until the water content reaches around 50–60%, worker bees engage in active evaporation behavior, which includes wing-fanning to promote circulation throughout the hive and sucking up, regurgitating, and retaining the nectar between their mandibles to expand its surface area. After that, it is put into wax cells and moved around a lot while passively draining until the water content reaches about 13–25%.

Honey contains over 200 distinct compounds, most of which are sugars, water, and other substances such as proteins (enzymes). Organic acids, minerals (calcium, copper, iron, magnesium, manganese, phosphorus, potassium, sodium, and zinc), vitamins (especially niacin, riboflavin, B6, thiamine, and pantothenic acid), pigments, phenolic compounds, and a variety of volatile compounds are among the materials extracted from the honey harvesting process.

The main ingredients in honey are carbohydrates. Glucose and fructose make up 75% of the monosaccharide sugars in natural honey, while sucrose and maltose are around 10–15%. The main variables affecting honey’s sugar content are climate, processing, storage, and place of origin (both botanical and geographical). Honey’s physicochemical characteristics, such as its viscosity, crystallization, thermal behavior, and viscoelastic behavior, are influenced by its sugar content. The honey composition is also related to its botanical and geographical origin, as well as its environmental and management conditions.

With at least 8000 distinct known structures, phenolic compounds, often known as polyphenols, are significant chemical groups found in a wide range of plants. Plants also generate them as a secondary metabolite. Some of these phenolic chemicals are found in the honey as well.

Based on their fundamental structural differences, phenolic compounds can be broadly classified into ten types: simple phenols, phenolic acids, coumarins and isocoumarins, naphthoquinones, xanthones, stilbenes, anthaquinones, flavonoids, and lignins. Of the polyphenolic classes, flavonoids are the most significant, with over 5000 substances having been identified to date. The naturally occurring antioxidants known as flavonoids have a variety of biological effects, such as antibacterial, anti-inflammatory, antiallergic, antithrombotic, and vasodilatory properties [9].

Numerous polyphenols have been identified in honey. Certain types of honey were identified by their polyphenol content; sunflower honey was identified by quercetin and rosemary honey by flavanol kaempferol. Chestnut honey has been discovered to include hydroxy-cinnamates, such as caffeic acid, ferulic acid, and p-coumaric acid. Most European honey samples also included propolis flavonoids, such as pinocembrin, pinobanksin, and chrysin, which are characteristic of the plant [10,11].

Manuka honey (a monofloral honey native to New Zealand), Tualang honey (a multifloral honey originating from Malaysia), and buckwheat honey (a monofloral honey derived from various geographical origins) have the highest content of flavonoids, while gelam and acacia honey have the lowest content [12].

Research has been performed on the anti-proliferative properties of many of the specific components of honey. A number of cancer cell lines have been shown to be susceptible to the harmful effects of chysin, one of the most thoroughly researched phenolics present in honey (with a concentration of about 0.2 mg per 100 g [13]). Chrysin, for instance, at concentrations of 25 µM and 50 µM was shown to inhibit human melanoma (A375) by 15% and 25% and to inhibit murine melanoma (B16-F1) cell lines by 10% and 20% after a 24 h treatment [14].

Quercetin (present in honey with a concentration of around 0.31 mg/100 g [15]) has been shown to have anti-proliferative effects on some cancer cell lines such as HL-60 leukemia, MCF-7 human breast cancer, Caco-2 human colon adenocarcinoma, and PC-3 and DU145 prostate cancers [16]. Quercetin was also found to have a synergistic anti-proliferative effect when combined with cisplatin [17].

Additional components of honey, such as luteolin, kaempferol, apigenin, and caffeic acid, have been investigated for their potential to prevent cancer cell proliferation [14,18].

Another promising active ingredient found in honey products is caffeic acid phenethyl ester (CAPE). Strong in vitro and in vivo inhibitory potentials have been observed for CAPE in a variety of cancer types, including breast, lung, colon, liver, pancreatic, melanoma, and gastric cancer [19,20].

It is noteworthy that, at concentrations 10- to 100-fold higher than those found in honey, each phenolic component individually inhibits the growth of cells [21]. This implies that either there are other cytotoxic components of honey that have not yet been investigated, or synergism between these substances results in anti-proliferative actions in honey even at low concentrations.

## 3. Honey’s Ability to Inhibit Cancer Cells

### 3.1. The Antiapoptotic Activity of Honey

Apoptosis is a type of controlled cell death that aids in controlling cell division and getting rid of unhealthy cells [22]. Cancer cells have several apoptotic mechanisms that are dysregulated, which promotes the cells’ survival and immortality.

Apoptosis is induced in many types of cancerous cells by honey through the depolarization of the mitochondrial membrane [23]. Due to its high tryptophan and phenolic content, honey is known to increase the activation level of caspase-3 and the cleavage of poly (ADP-ribose) polymerase (PARP) in human colon cancer cell lines [24].

Moreover, honey produces reactive oxygen species (ROS), which activate p53. The phosphorylation of pro- and antiapoptotic proteins, including as Bcl-2 and Bax, is then modulated by p53 [25].

Wistar rats treated with honey as an adjuvant therapy with *Aloe vera* exhibit increased expression of the proapoptotic protein Bax and decreased expression of the antiapoptotic protein Bcl-2 [26]. By triggering caspase-9, which in turn triggers caspase-3, the executioner enzyme, manuka honey induces apoptosis in cancerous cells. In addition, DNA breakage, PARP activation, and decreased Bcl-2 expression are all involved in manuka-induced apoptosis [27]. So, the results show that manuka honey, which is effective at concentrations as low as 0.6% (*w*/*v*) at 24 h, has a strong anti-proliferative effect on all cancer cell lines in a time- and dose-dependent manner.

### 3.2. The Anti-Proliferative Properties of Honey

Over the course of life, epithelial cells divide. The G1, S, G2, and M phases are the distinct stages that make up the cell cycle. Many distinct proteins control every step of the cell cycle. Cyclin-dependent kinases and cyclins themselves make up the cell cycle control panel. A cell’s fate is determined at the G1/S phase transition, which is a crucial regulatory point [28]. Cells can choose to undergo quiescence, proliferation, differentiation, or death. Tumorigenesis is associated with the overexpression and deregulation of cell cycle growth factors, including cyclin D1 and cyclin-dependent kinases (CDKs) [28]. Cancer is characterized by disrupted regulation. Ki-67, a nuclear protein, can be used to measure the “growth fraction” in cell proliferation. It is not present in the quiescent phase (G0) but is expressed throughout the cell cycle (G1, S, G2, and mitosis) [29].

There is evidence that honey influences cell cycle arrest [30]. When rats were given honey along with *Aloe vera* solution (670 µL/kg dose of the *Aloe vera* and honey solution daily), their tumor cells’ production of Ki-67 was significantly reduced [14]. This implies that by stopping the cell cycle, honey treatment may reduce the growth of malignant cells [26].

It has been shown that honey and some of its constituents (such as flavonoids and phenols) inhibit the cell cycle of tumor cell lines in the G0/G1 phase, including colon, glioma, and melanoma [31]. Following the downregulation of numerous biological processes via tyrosine cyclooxygenase, ornithine decarboxylase, and kinase, there is an inhibitory effect on the growth of neoplastic cells.

So, the disruption of the cell cycle caused by honey or its constituents inhibits the development of cells [32]. Additionally, p53 controls the cell cycle, which is crucial for tumor suppression. According to reports, honey affects how p53 regulation is modulated [24].

### 3.3. Anti-Inflammatory Role of Honey

There is a link between chronic inflammatory diseases and a higher risk of cancer [33]. The inflammatory process that is often triggered in malignancies involves two crucial elements: the nuclear factor kappa B (NF-kB) and mitogen-activated protein kinase (MAPK) pathways [34]. The production of several inflammatory proteins and genes, such as cyclooxygenase-2 (COX-2), C-reactive protein (CRP), and lipoxygenase-2 (LOX-2), as well as pro-inflammatory mediators or cytokines including interleukin 1 (IL-1), IL-6, and tumor necrosis factor-α (TNF-α), is the consequence of the activation of MAPK and/or NF-κB. It is well recognized that angiogenesis and the inflammatory-related etiology of cancer are significantly influenced by all of these pathways and pro-inflammatory intermediaries [35].

Recent research by Batumalaie et al. demonstrated that in HIT-T15 cells, which are cultured Syrian hamster islet cells transformed with SV40 virus, gelam honey treatment (at concentrations of 20, 40, 60, and 80 µg/mL) decreased the expression of NF-κB and MAPK with a maximum inhibition at 20 µg/mL [36]. It has also been demonstrated that chrysin, present in honey, can cause apoptosis in B16-F1 and A375 melanoma cells by altering MAPK [37]. Chrysin was also shown to increase apoptosis in cancer cell lines by affecting TNF-related apoptosis-inducing ligand (TRAIL) [38]. Additionally, leukocyte (monocyte and neutrophil) infiltration and edema were decreased by honey [39]. In vivo research has also produced data recently supporting honey’s anti-inflammatory properties. Rats with inflammation caused by carrageenan were used in a study by Hussein et al. to assess the anti-inflammatory properties of honey (1 or 2 g/kg, per os of Malaysian gelam honey) [40]. It has also been observed that honey’s cytotoxic action (in particular, of Tualang honey) on non-small cell lung cancer (NSCLC) cells is partially mediated through the regulation of inflammatory cytokines [41].

Tualang honey is a multiflora rainforest honey originating from Malaysia. The *Apis dorsata* wild giant bees, who establish their colony atop the Tualang tree (*Koompassia excelsa*), are the producers of this honey. Because of its high concentration of polyphenolic chemicals, Tualang honey naturally displays a dark brown color. Compared to manuka honey, the polyphenolic compounds of Tualang honey have significantly higher concentrations of phenolic acid and flavonoids.

It has also been observed that honey’s cytotoxic action on NSCLC cells is partially mediated through the regulation of inflammatory cytokines [41]. These results imply that honey’s anti-inflammatory properties, which are mediated via inhibiting NF-κB and MAPK signaling pathways and attenuating pro-inflammatory mediators, may significantly contribute to honey’s anticancer properties. Honey’s phenolic components and flavonoids are the main components responsible for its anti-inflammatory properties [42,43].

### 3.4. The Impact of Honey on Tumor Necrosis Factor (TNF)

Tumor necrosis factor (TNF) plays a role in cancer genesis and progression [44]. TNF activates NF-kB, which causes inflammation and is linked to various disorders [45].

Inducible nitric oxide synthase (iNOS), chemokines, LOX-2, and COX-2 are among the inflammatory genes that are expressed as a result of the activation of NF-kB [45]. For many tumor cells, it functions as a growth factor. In contrast, TNF-α has also been demonstrated to play a crucial role as a cytokine in host defense mechanisms. It has been demonstrated to have both positive and negative effects on the development or prevention of infectious disorders [46].

Apalbumin-1 and apalbumin-2, two proteins found in royal jelly (RJ) and honey, showed anticancer effects. These proteins are able to induce the release of TNF-α, IL-1, and IL-6 from macrophages [47]. At concentrations of 1% *w*/*v*, pasture, jelly bush, and manuka honeys activate monocytes to produce IL-1β, IL-6, and TNF-α [48].

One potential mechanism for controlling apoptosis and inflammation through these cytokines is the binding of TNF-receptor (TNF-R) to TNF-α and adaptor proteins, such as TNF-R-associated death domain protein (TRADD), TNF receptor-associated factor (TRAF), and receptor-interacting protein (RIP) [49]. This production of TNF-α can be crucial in controlling essential cellular processes such inflammation, cell division, and apoptosis as a major cytokine [50].

### 3.5. Honey’s Ability to Modulate Oxidative Stress

It is still debatable how oxidative stress and ROS affect the development and suppression of cancer. There is proof that ROS have both stimulatory and inhibitory effects on cancer. Cell growth is aided by low ROS levels [51]. However, oxidative damage is caused by elevated ROS levels, which have been linked to several cancer types [52].

Recent research suggests that cancer cells may perish when exposed to high levels of ROS and/or lipid peroxidation products, despite their conflicting effects [53].

Strong antioxidants and scavengers of free radicals are found in honey [54]. Its antioxidant qualities are thought to be responsible for a number of its biological actions [55]. Thus, honey’s potential to prevent cancer growth and proliferation may be partially mediated through its ability to modulate oxidative stress.

Improved antioxidant status was linked to the anti-neoplastic actions of *Nigella sativa* and honey (pure unfractionated honey obtained from bees fed on Egyptian clover flowers) on hepatocellular carcinoma cells, according to data from Hassan and colleagues’ study [56]. Conversely, it has been demonstrated by Jaganathan and Mandal that honey’s anticancer impact is mediated by elevated oxidative stress. According to the study, the colon cancer cells HCT-15 and HT-29 experienced non-protein thiol depletion as a result of honey (obtained from Kharagpur (West-Bengal state, India)) treatment. It has also been demonstrated that the anti-proliferative properties of honey are linked to elevated ROS production [25].

Honey’s dual actions on cancer cells are most likely caused by the phenolic compounds found in it. In addition to being antioxidants, phenolic substances are readily oxidized. Honey’s phenolic compounds have the potential to release ROS that damage DNA and cause cell death through their interactions with transition metal ions such as copper [57,58]. It has been shown that a number of flavonoids and phenolics can cause ROS production in cancer cells [59].

Martinotti et al. [60] explored the cytotoxicity of three varieties of honeys (i.e., manuka, buckwheat, and acacia honey), distinguished by varying quantities of polyphenolic chemicals, using A431, a cancer cell line derived from an epidermal carcinoma of the vulva. They also explain how manuka honey was the most effective honey affecting the intracellular ROS and intracellular calcium homeostatic balance, which induces the cancer cell line to undergo apoptosis and death. This represents the first plausible method by which honey could cause an intracellular ROS increase and the homeostatic balance of [Ca^2+^]_i_ to change, ultimately causing apoptosis in cancer cells (see also Figure 1). Manuka honey’s capacity to sustain a high H_2_O_2_ permeability appears to improve this mechanism. Therefore, a novel tandem mechanism of channel (aquaporin-3) gating in conjunction with H_2_O_2_-mediated apoptosis may be responsible for the possible anticancer action of manuka honey.

### 3.6. Honey’s Impact on Cancer Formation and Growth

Several studies have been carried out to investigate the potential role of honey in halting the growth and development of tumors. The majority of these studies use in vitro methods, and only a small number use in vivo models.

Globally, breast cancer is the leading cause of cancer-related deaths. About 12% of women are expected to get breast cancer at some point in their lives [1]. In addition to various other variables, the levels of circulating estrogens and the dysregulation of estrogen signaling pathways are the main factors influencing the development and progression of breast cancer [61]. Consequently, the estrogen receptor (ER) signaling pathway is frequently the focus of breast cancer therapy. The potential impact of honey on this important pathway has been examined in a number of ways.

Tsiapara et al. [62] assessed the efficacy of honey extracts from pine, fir, and Greek thyme to regulate the estrogenic viability and activity of breast cancer cells (MCF-7). Samples of honey were gathered on the islands of Euboea (Central East Greece), Karpenisi (North-West Greece), and Crete (Iraklion, South Greece). The researchers discovered that, depending on the concentration (0.2–125 μg/mL honey extracts for 48 h), the honey samples had a biphasic effect on MCF-7 cells, acting as an estrogen at high concentrations and an antiestrogenic agent at low ones. The thyme and pine honey extracts were found to counteract estrogen activity in the presence of estradiol, but fir honey extract increased estrogen activity in MCF-7 cells. Variations in the impact of the three honey extracts on cell viability were also identified in the study. The study found that pine and thyme honey had no effect on MCF-7 cells, while fir honey boosted cell viability. The honey extracts’ dual effects are likely due to their high phenolic content, including kaempferol and quercetin. Phytoestrogens, also known as phenolic substances, can have both stimulating and inhibiting effects [63].

The human breast cancer cell lines MDA-MB-231 and MCF-7 have also been used to illustrate the cytotoxic impact of Tualang honey [23]. Increased lactate dehydrogenase (LDH) leakage from the cell membranes indicated the cytotoxicity. It has been demonstrated that Tualang honey (1–10% for up to 72 h) lowers mitochondrial membrane potential and causes apoptosis. Additionally, the researchers discovered that honey had no cytotoxic effects on MCF-10A cells, i.e., a normal breast cell line. This implies that Tualang honey’s cytotoxic effects are limited to breast cancer cell types. This is significant because an effective chemotherapeutic drug must possess both selectivity and specificity.

The anti-proliferative, anti-metastatic and anticancer properties of honey on rodent breast tumors have also been verified by numerous studies. Honey has been shown to have anti-metastatic properties in a mouse model of breast cancer when treated prior to tumor cell injection with a preparation given to mice intraperitoneally at a dose of 100 mg kg^−1^ body weight for three consecutive days [32].

Honey’s ability to suppress the metastatic potential of human breast cancer cells may be attributed to its flavonoids, such as chrysin [64]. In a similar way, research examined the anticancer effect of two honey samples—one with a higher phenolic content than the other—on solid carcinoma and Ehrlich ascites. Ehrlich ascites carcinoma growth was found to be significantly inhibited by both types of honey, administered to each mouse intraperitoneally daily; however, the honey with the higher phenolic content had a stronger anticancer impact [65].

The possible anti-tumoral effects of manuka honey (MH) on estrogen receptor-positive and -negative breast cancer were studied by Marquez-Garban and coworkers [66]. They discovered, both in vitro and in vivo, that manuka honey (with concentrations ranging from 0.3 to 5.0% (*w*/*v*)) suppress cell growth in a dose-dependent way.

The nectar that honey bees gather from pollinating Manuka tea trees (*Leptospermum scoparium)* is used to make MH, a monofloral honey. It is well recognized that MH possesses antibacterial, antioxidant, and tissue-protective/healing properties [67]. Methylglyoxal, the primary ingredient thought to be responsible for MH’s antibacterial and antioxidant qualities, is present in high proportions in its unique composition [67].

The authors highlighted the signaling pathways that might be involved in the mechanisms of action of MH. The MH anticancer therapeutic activity appears to be mediated by AMP kinase (AMPK) activation, the suppression of downstream mTOR (mammalian target of rapamycin) signaling, and STAT3 (signal transducer and activator of transcription 3).

In particular, the in vitro growth of MCF7 cells, an extensively utilized breast adenocarcinoma-derived epithelial cancer cell line, was dose-dependently suppressed by MH, which also activated PARP to cause apoptosis. Moreover, MH induced AMPK and suppressed STAT3 and the downstream signaling of mTOR.

It is noteworthy that MH taken orally prevented MCF7 tumor xenografts from growing in vivo while having little adverse effects. These results suggest that natural compounds with strong antitumor activity and selectivity against hormone receptor-positive breast cancers, such as manuka honey, could be developed further as a supplement or as a possible substitute for cytotoxic anticancer medications that have more non-selective side effects.

Liver cancer is becoming more common, just as other cancers are. Among liver cancers, hepatocellular carcinoma (HCC) is the most common. Numerous variables, including infection with the hepatitis C or hepatitis B virus, obesity, diabetes, and inherited and societal risk factors such excessive alcohol intake, are associated with an increased incidence of HCC [68]. Despite significant advancements in HCC treatment, patients’ quality of life is still quite low. This implies that we should never stop trying to develop more effective treatments.

Numerous recent research has demonstrated the anticancer benefits of honey on liver cancer cells. When honey was applied to human hepatocellular carcinoma (HepG2) cells, Hassan et al. found that while it increased the total antioxidant status (TAS), it significantly decreased the number of viable HepG2 cells and nitric oxide (NO) levels [56]. ROS may be responsible for the HepG2 cell line viability or survival, based on these findings. Cell growth, division, and proliferation are all aided by moderate ROS levels. This perspective is supported by the lower NO levels observed after honey therapy. This study demonstrates that honey will always increase TAS by scavenging ROS.

Moreover, the IC_50_ value of gelam honey towards HepG2 was found to be 25% in a study that examined the anti-proliferative effects of the honey on HepG2, while it was 70% for normal human hepatocytes [30]. This demonstrates that liver cancer cells are specifically sensitive to gelam honey.

Few studies have been conducted to investigate honey’s possible ability to prevent or inhibit the in vivo development of liver tumors. In rats, diethylnitrosamine (DEN)-induced hepatic cancer was studied in relation to honey’s impact on the tumor’s growth and progression [69]. Rats, with any treatment for six months, had a variety of abnormalities in their livers. The liver of honey-treated DEN-induced rats showed a significant reduction in several abnormalities. The results of this study indicate that honey protects rats from chemically induced hepatocarcinogenesis and has anticancer effects on liver cancer cells [69].

Colon cancer ranks fourth in terms of causes of mortality globally and is the third most common type of cancer worldwide [70]. It has been determined that environmental variables are mostly responsible for the pathophysiology of colorectal cancer [71]. There is an urgent need to supplement the present approaches to this neoplasm since, despite advancements in treatment, including postoperative care, recurrence and mortality rates remain high [72]. The colon cancer cell line HT 29 was used in a study to examine the chemopreventive effects of gelam and nenas monofloral honeys. The results indicated that the honey samples reduced colon cancer cell multiplication. Additionally, both honeys inhibited inflammation in H_2_O_2_-induced colon cancer cells and damaged DNA in a dose-dependent way [73]. Moreover, cytotoxicity assays demonstrated that gelam honey was more successful in inhibiting the growth of colon cancer cells than nenas monofloral honey, with IC_50_ values of 39.0 mg/mL and 85.5 mg/mL, respectively, following a 24 h treatment period.

In a recent work, the colon cancer cell lines HT 29 and HCT 15 were used to test the apoptotic effect of several crude honey samples [25]. The research verified the previously documented anti-proliferative impact of honey on colon cancer cells and additionally showed that this effect was correlated to the phenolic content. The anti-proliferative impact against colon cancer cells increases with the amount of phenolic content [24]. Honey not only had an anti-proliferative impact but also caused intracellular non-protein thiols to be depleted, which in turn caused apoptosis. Additionally, it raised the production of ROS and decreased the potential of the mitochondrial membrane [25].

The bioactivity of three Greek honeys—pine, fir, and thyme honey—on the PC-3 cell line, i.e., prostate cancer cells, was examined in a study that revealed notable variations in the honeys’ effects on cellular growth [62]. The results demonstrated that only thyme honey (at final concentrations ranging from 0.2 to 125 μg/mL for a 48 h incubation) significantly decreased the PC-3 cell viability; extracts from pine and fir honey had no such impact. These results imply that honey possess an anti-proliferative effect on neoplastic cells of the prostate.

### 3.7. Honey’s Impact on Cancer Stem Cells

Chemoresistance is closely linked to cancer stem cells (CSCs) in different forms of cancer. Self-renewal is one of the essential characteristics of CSCs, and it is thought that CSCs may have a role in tumor metastases and recurrences as well as in resistance to radiation and chemotherapy. In this regard, manuka honey has been shown in recent years to have a pleiotropic effect on CRC cells; this effect is thought to be attributed to the honey’s abundance of bioactive chemicals. In detail, it was found that manuka honey could reduce cell viability and proliferation, weaken the metabolic pathway and mitochondrial bioenergetic function, and block some crucial stages of cell migration and metastasis, all of which increased the impact of a chemotherapeutic drug such as 5-FU (fluorouracil) [74].

Cianciosi and colleagues [75] demonstrated that manuka honey (UMF, Unique Manuka Factor, 15+, for 48 h) shows a chemosensitizing impact towards 5-FU on colon CSC-like cells generated from the HCT-116 cell line. Instead of using a standard two-dimensional colonosphere experimental model, the scientists used a three-dimensional model that more closely resembles the in vivo environment. In particular, manuka honey was linked to CSCs’ diminished capacity for self-renewal, controlling the expression of many genes involved in the Wnt/β-catenin, Hedgehog, and Notch pathways.

Even though these are preliminary findings, they are significant because they imply that manuka honey and 5-FU could function in concert, opening up new avenues for characterizing natural substances in addition to conventional chemotherapeutic approaches and reducing their side effects.

## 4. Conclusions

Honey may be an anticancer agent through a number of pathways, according to mounting evidence suggesting that honey is not cytotoxic to normal cells, but it is strongly and specifically cytotoxic to tumor cells.

Research has shown that honey affects several cell signaling pathways, including those that result in apoptosis, anti-proliferative, and anti-inflammatory effects, even if the precise mechanism is yet unknown. There are still many unanswered questions. Furthermore, honey that comes from various floral sources may have distinct effects.

To increase our knowledge of honey’s protective effects against cancer, more research is required. Humans may not experience the same outcomes obtained from animal trials or cell culture experiments.

Before suggesting honey’s use in therapeutic treatments for cancer patients, more research should be performed to validate its anticancer properties. To verify the legitimacy of honey as a stand-alone treatment or as an adjuvant, prospective randomized controlled clinical trials are strongly encouraged.

## Figures and Tables

**Figure 1 ijms-25-11724-f001:**
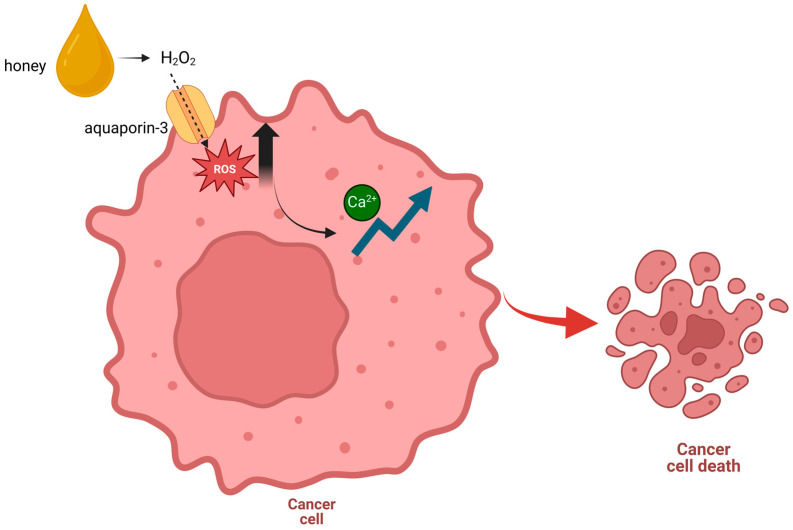
Honey’s molecular mechanism on cancer cells, as described by Martinotti et al. [60]. In the extracellular space, honey causes the production of H_2_O_2_ at the micromolar level. Through aquaporin-3, H_2_O_2_ crosses the plasma membrane and initiates intracellular reactions mediated by an [Ca^2+^]_i_ increase (Created in BioRender. https://biorender.com/f28l375 (accessed on 25 September 2024).

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
