# Peer review of "Understanding the Anticancer Properties of Honey"

_ijms, 2024, doi:10.3390/ijms252111724_

Round 1
Reviewer 1 Report
Comments and Suggestions for Authors
The authors present a narrative review of a topic that is of interest and is within the preferred themes of this journal. However, this manuscript is too brief, it presents the topics in a simplistic and unscientific or academic way. The authors do not adequately describe the mechanisms or the context of the disease. No specific therapies or preclinical models are described. This manuscript is a mini-review. I propose that it be rejected and that the authors make an effort to create a solid review.
Comments on the Quality of English LanguageThe English is very difficult to understand/incomprehensible.
Author Response
The authors present a narrative review of a topic that is of interest and is within the preferred themes of this journal. However, this manuscript is too brief, it presents the topics in a simplistic and unscientific or academic way. The authors do not adequately describe the mechanisms or the context of the disease. No specific therapies or preclinical models are described. This manuscript is a mini-review. I propose that it be rejected and that the authors make an effort to create a solid review.
Comments on the Quality of English Language. The English is very difficult to understand/incomprehensible.
We consider this referee's comments not only useless, but absolutely offensive towards our work. Having accused the manuscript of being written in a “unscientific or academic way” invites us not to consider his/her position, which we believe to be in bad faith.
Moreover, the statement that “The authors do not adequately describe the mechanisms or the context of the disease” leads us to think that the author has not read the manuscript, not understanding why we should extensively describe what cancer is or how it develops, which is outside the topic of this review.
“No specific therapies or preclinical models are described”. What type of preclinical model should there be described? Throughout the text, when we talk about in vivo experiments or animal models, we have mentioned the models used and the type/concentration of honey utilized.
The scope of our review is to demonstrate that honey is able to affect a number of cell-signaling pathways, including those that cause apoptosis, anti-proliferative and anti-inflammatory effects. However, many questions remain unresolved as well as the effects of honey of different botanical origin that can be different. In addition, the results obtained in vitro and in vivo studies cannot be directly transferred to humans. Prospective randomized controlled clinical trials are strongly needed to understand the real potential of honey.
Among other things, we had positive and enthusiastic comments from two other referees. Referee’s comments that allowed us to improve the manuscript.
Reviewer 2 Report
Comments and Suggestions for Authors
This is an interesting paper dealing with the current knowledge on honey and its possible impact on cancer based on the available literature on in vitro and in vivo studies.
Honey is considered as a possible tool for complementary and alternative medicine given that honey is not cytotoxic to normal cells but is highly and particularly cytotoxic to tumor cells.
Research has demonstrated that honey affects a number of cell-signaling pathways, including those that cause apoptosis, anti-proliferative and anti-inflammatory effects. However, many questions remain unresolved as well as the effects of honey of different botanical origin that can be different. In addition, the results obtained in vitro and in vivo studies cannot be directly transferred to humans. Prospective randomized controlled clinical trials are strongly needed.
Line 47: revise the 2 sentences
Line 270: please check this statement about production of Manuka honey all over the world

Author Response
This is an interesting paper dealing with the current knowledge on honey and its possible impact on cancer based on the available literature on in vitro and in vivo studies.
Honey is considered as a possible tool for complementary and alternative medicine given that honey is not cytotoxic to normal cells but is highly and particularly cytotoxic to tumor cells.
Research has demonstrated that honey affects a number of cell-signaling pathways, including those that cause apoptosis, anti-proliferative and anti-inflammatory effects. However, many questions remain unresolved as well as the effects of honey of different botanical origin that can be different. In addition, the results obtained in vitro and in vivo studies cannot be directly transferred to humans. Prospective randomized controlled clinical trials are strongly needed.
We appreciate the reviewer's good assessment of the manuscript.
Line 47: revise the 2 sentences
We have modified these two sentences
Line 270: please check this statement about production of Manuka honey all over the world
In fact, manuka honey is a monofloral honey produced from the nectar of the manuka tree, Leptospermum scoparium. The manuka tree is indigenous to New Zealand and some parts of coastal Australia, but manuka honey is today produced globally.
The natural habitat of manuka can be described as lowland to subalpine areas in various habitats, especially open slopes, river banks, forest margins, and scrub, where it often forms the dominant vegetation. Although native to New Zealand, the manuka plant has spread to Australia, the United States, several European countries, Israel and South Africa.
We have amended this sentence.
English language fine. No issues detected.
Thank you
Reviewer 3 Report
Comments and Suggestions for Authors
Thank you for submitting the manuscript "Honey and cancer: the multifaceted impact of different honey types" to IJMS. Overall, the manuscript is superficial and needs to be improved. The authors need to describe in depth the studies found in this literature review, since this is a food that is being indicated as an adjuvant to a treatment to demonstrate its functional potential. In addition, I have other small questions.
Line #27: From the way it is, it seems that traditional and alternative medicine can be used to replace already established cancer treatments. This is not true. In fact, CAM can be used as a strategy and adjuvant to treatment, but this replacement should never be recommended or stated so openly in a scientific manuscript.
Line #41: There are numerous studies that, in addition to identifying the composition of honey, also quantified it. I am very surprised that the authors write a review article without citing the quantities of important active compounds so that it can later be associated with the biological function. My suggestion is to use a good number of references and provide the average values ​​found by them. Line#86: the response dose and experimental models must be indicated in the entire subitem.
Line#104: the text is superficial since the response dose values ​​and treatment duration are not mentioned.
Comments on the Quality of English LanguageEnglish is not good.
Author Response
Thank you for submitting the manuscript "Honey and cancer: the multifaceted impact of different honey types" to IJMS. Overall, the manuscript is superficial and needs to be improved. The authors need to describe in depth the studies found in this literature review, since this is a food that is being indicated as an adjuvant to a treatment to demonstrate its functional potential. In addition, I have other small questions.
We appreciate the reviewer's time in assessing our work and making suggestions about how to make the ms better.
Line #27: From the way it is, it seems that traditional and alternative medicine can be used to replace already established cancer treatments. This is not true. In fact, CAM can be used as a strategy and adjuvant to treatment, but this replacement should never be recommended or stated so openly in a scientific manuscript.
We thank reviewer for this suggestions and for the opportunity to better define this concept. We do not want to suggest to replace classic anticancer treatments using complementary and alternative medicine (CAM) or much less with honey.
However, it is undoubtable that the use of CAM appears to be increasing worldwide and is rapidly evolving and growing in the healthcare industry.
This emphasizes the need for its regulation, with a focus on training and research.
So, we reviewed research demonstrating that honey is able to affect a number of cell-signaling pathways, including those that cause apoptosis, anti-proliferative and anti-inflammatory effects. However, many questions remain unresolved as well as the effects of honey of different botanical origin that can be different. In addition, the results obtained in vitro and in vivo studies cannot be directly transferred to humans. Prospective randomized controlled clinical trials are strongly needed to understand the real potential of CAM and in particular of honey.
We have amended these sentences in order to avoid any misunderstanding.
Line #41: There are numerous studies that, in addition to identifying the composition of honey, also quantified it. I am very surprised that the authors write a review article without citing the quantities of important active compounds so that it can later be associated with the biological function. My suggestion is to use a good number of references and provide the average values ​​found by them.
We appreciate the reviewer's suggestion. The active ingredients in honey have been introduced and their recognized biological effects have been reviewed.
Line#86: the response dose and experimental models must be indicated in the entire subitem.
We thank reviewer for this suggestion. We have inserted information concerning the concentration and type of honey utilized as well as the duration of treatment, when these details are available in the original manuscripts.
Line#104: the text is superficial since the response dose values ​​and treatment duration are not mentioned.
We thank reviewer for this suggestion. We have amended this sentence.
Comments on the Quality of English Language: English is not good.
The ms has undergone significant revisions.
Round 2
Reviewer 3 Report
Comments and Suggestions for Authors
This reviewer is grateful for the corrections made by these authors. However, in my opinion, many things still need to be revised, as it still seems to me to be a biased work from the point of view that it needs to be made clear at the end of the entire text what the next steps of the research are to show that there is in fact some potential for honey from a health point of view. I think that this "beautification" of foods to make them "functional" should be carried out with great caution by scientists so as not to transform unbiased scientific information into media information that will make the population follow the nutraceutical point of view, but still with weak evidence (since there are no studies in animals or humans). This needs to be very clear and I still do not feel comfortable accepting the manuscript for this reason. Some small points (but there are many others):
- The title is the first problem in my opinion, as it has already demonstrated that it can lead to erroneous interpretations. - Line#56: In my opinion, statements like the one in line#56, for example, weaken the work by indicating that the authors cannot demonstrate what biochemical transformations compounds collected by bees undergo/undergo to become honey. And this does need to be explained, even if in general terms.
- Line#73: not only by the sugar content, but also by other components that are in the honey and also by factors external to the honey such as collection (type), temperature, etc.
- Line#159: compared to which treatment? control or A. vera?
- Line#182: at what concentration was it effective?
- Line#229: no concentration was added to "justify" the action of honey in this type of disease. The same happened with line#278.
Comments on the Quality of English LanguageModerate editing of English language required.
Author Response
This reviewer is grateful for the corrections made by these authors. However, in my opinion, many things still need to be revised, as it still seems to me to be a biased work from the point of view that it needs to be made clear at the end of the entire text what the next steps of the research are to show that there is in fact some potential for honey from a health point of view. I think that this "beautification" of foods to make them "functional" should be carried out with great caution by scientists so as not to transform unbiased scientific information into media information that will make the population follow the nutraceutical point of view, but still with weak evidence (since there are no studies in animals or humans). This needs to be very clear and I still do not feel comfortable accepting the manuscript for this reason.
We appreciate the reviewer's time in assessing our work and making suggestions about how to make the ms better.
Further studies should be conducted to confirm honey's anticancer qualities before recommending its usage in therapeutic treatments for cancer patients. This idea has been now emphasized in the ms.
Some small points (but there are many others):
- The title is the first problem in my opinion, as it has already demonstrated that it can lead to erroneous interpretations.
To prevent any misunderstandings, we have changed the manuscript's title.
- Line#56: In my opinion, statements like the one in line#56, for example, weaken the work by indicating that the authors cannot demonstrate what biochemical transformations compounds collected by bees undergo/undergo to become honey. And this does need to be explained, even if in general terms.
We apologize for not being able to explain the bees' method of producing honey in the clearest, most effective manner. In order to prevent oversimplifying the process, we have added a few phrases to the text.
- Line#73: not only by the sugar content, but also by other components that are in the honey and also by factors external to the honey such as collection (type), temperature, etc.
The reviewer is right. We have modified this sentence
- Line#159: compared to which treatment? control or A. vera?
This study verified the influence of Aloe vera and honey on tumor growth and in the apoptosis process by assessing tumour size, the cell proliferation rate (Ki67) and Bax/Bcl-2 expression at 7, 14 and 20 days after Walker 256 carcinoma implant in Wistar rats distributed into two groups: the WA group – tumour-bearing rats that received a gavage with a 670 µL/kg dose of Aloe vera and honey solution daily, and the control group, tumour-bearing rats, which received only a 0.9% NaCl solution.
- Line#182: at what concentration was it effective?
The most effective concentration was 20 μg/mL.
In particular, HIT-T15 cells were pretreated with Gelam honey extract at concentrations of 20, 40, 60, 80 μg/mL for 24 hours, following which they were cultured with 20 mM or 50 mM glucose to determine the phosphorylation of NF-κB.
The phosphorylation of NF-κB was increased in HIT-T15 cells treated with 20 mM and 50 mM glucose alone as compared with control. Pretreatment with Gelam honey extract showed a 50% and 56% decrease (P < 0.05) (at 20 and 40 μg/ml) in the expression of phosphorylated NF-κB.
We have made improvements to the ms text.
- Line#229: no concentration was added to "justify" the action of honey in this type of disease. The same happened with line#278.
For line#278, it is difficult to indicate a concentration of honey utilized to study its effects against cancer cell lines. Authors have utilized honey of different origin, some authors have indications no honey characteristics, and other authors have utilized different preparations of honey (e.g. extraction from honey of several compounds), while other authors utilized honey in combination with other chemicals or drugs.
It is challenging to specify the concentration of honey used in the investigation of its anti-cancer cell line effects for line #278. Different types of honey have been used by authors; some authors did not indicate honey characteristics; some have used different honey preparations (such as extracting multiple compounds from honey); still others have combined honey with other substances, medications or drugs.
For line#229, it is possible that the referee mentioned incorrecly the line to which she/he refers. Specifically, at line 229, in the version after the first revision of the manuscript, the role of ROS is generically mentioned, but it will be difficult to “to "justify" the action of honey in this type of disease” as requested. We remain available to modify and implement the manuscript where necessary.
Moderate editing of English language required.
We have made improvements to the manuscript.
Round 3
Reviewer 3 Report
Comments and Suggestions for Authors
This reviewer thanks the authors for making all suggested corrections. In my opinion, the manuscript can be accepted in its current form.